# Unsupervised Discovery of Steerable Factors When Graph Deep Generative Models Are Entangled

**Shengchao Liu**  *shengchao.liu@umontreal.ca*
*Quebec AI Institute (Mila)*
*University de Montréal*

**Chengpeng Wang**  *cw83@illinois.edu*
*University of Illinois Urbana-Champaign*

**Jiarui Lu**  *jiarui.lu@umontreal.ca*
*Quebec AI Institute (Mila)*
*University de Montréal*

**Weili Nie**  *wnie@nvidia.com*
*Nvidia Research*

**Hanchen Wang**  *hw501@cam.ac.uk*
*University of Cambridge*

**Zhuoxinran Li**  *zhuoxinran.li@mail.utoronto.ca*
*University of Toronto*

**Bolei Zhou**  *bolei@cs.ucla.edu*
*University of California, Los Angeles*

**Jian Tang**  *jian.tang@hec.ca*
*Quebec AI Institute (Mila)*
*HEC Montréal*

*Reviewed on OpenReview:* *https://openreview.net/forum?id=wyU3Q4gahM*

## Abstract

Deep generative models (DGMs) have been widely developed for graph data. However, much less investigation has been carried out on understanding the latent space of such pretrained graph DGMs. These understandings possess the potential to provide constructive guidelines for crucial tasks, such as graph controllable generation. Thus in this work, we are interested in studying this problem and propose GraphCG, a method for the unsupervised discovery of steerable factors in the latent space of pretrained graph DGMs. We first examine the representation space of three pretrained graph DGMs with six disentanglement metrics, and we observe that the pretrained representation space is entangled. Motivated by this observation, GraphCG learns the steerable factors via maximizing the mutual information between semantic-rich directions, where the controlled graph moving along the same direction will share the same steerable factors. We quantitatively verify that GraphCG outperforms four competitive baselines on two graph DGMs pretrained on two molecule datasets. Additionally, we qualitatively illustrate seven steerable factors learned by GraphCG on five pretrained DGMs over five graph datasets, including two for molecules and three for point clouds.

# 1 Introduction

The graph is a general format for many real-world data. For instance, molecules can be treated as graphs (Duvenaud et al., 2015; Gilmer et al., 2017) where the chemical atoms and bonds correspond to the topological nodes and edges respectively. Processing point clouds as graphs is also a popular strategy (Shi & Rajkumar, 2020; Wang et al., 2020), where points are viewed as nodes and edges are built among the nearest neighbors. Many existing works on deep generative models (DGMs) focus on modeling the graph data and improving the synthesis quality. However, understanding the pretrained graph DGMs and their learned representations has been much less explored. This may hinder the development of important applications like *graph controllable generation* (also referred to as *graph editing*) and the discovery of interpretable graph structure.

Concretely, the graph controllable generation task refers to modifying the steerable factors of the graph so as to obtain graphs with desired properties easily (Drews, 2000; Pritch et al., 2009). This is an important task in many applications, but traditional methods (*e.g.*, manual editing) possess inherent limitations under particular circumstances. A typical example is molecule editing: it aims at modifying the substructures of molecules (Mihalić & Trinajstić, 1992) and is related to certain key tactics in drug discovery like functional group change (Ertl et al., 2020) and scaffold hopping (Böhm et al., 2004; Hu et al., 2017). This is a routine task in pharmaceutical companies, yet, relying on domain experts for manual editing can be subjective or biased (Drews, 2000; Gomez, 2018). Different from previous works, this paper starts to explore unsupervised graph editing on pretrained DGMs. It can act as a complementary module to conventional methods and bring many crucial benefits: (1) It enables efficient graph editing in a large-scale setting. (2) It alleviates the requirements for extensive domain knowledge for factor change labeling. (3) It provides a constructive perspective for editing preference, which can reduce biases from the domain experts.

**Disentanglement for editing.** One core property relevant to the general unsupervised data editing using DGMs is disentanglement. While there does not exist a widely-accepted definition of disentanglement, the key intuition (Locatello et al., 2019) is that a disentangled representation should separate the distinct, informative, and steerable factors of variations in the data. Thus, the controllable generation task would become trivial with the disentangled DGMs as the backbone. Such a disentanglement assumption has been widely used in generative modeling on the image data, *e.g.*, $\beta$-VAE (Higgins et al., 2017) learns disentangled representation by forcing the representation to be close to an isotropic unit Gaussian. However, it may introduce extra constraints on the formulations and expressiveness of DGMs (Higgins et al., 2017; Ridgeway & Mozer, 2018; Eastwood & Williams, 2018; Wu et al., 2021).

**Entanglement on pretrained graph DGMs.** Thus for graph data, one crucial question arises: *Is the latent representation space from pretrained graph DGMs disentangled or not?* In image generation, a series of work (Collins et al., 2020; Shen et al., 2020a; Härkönen et al., 2020; Tewari et al., 2020; Wu et al., 2021) has shown the disentanglement properties on pretrained DGMs. However, such property of pretrained graph DGMs is much less explored. In Section 3, we first study the latent space of three pretrained graph DGMs and empirically illustrate that the learned space is not perfectly disentangled or entangled. In what follows, we adopt the term "entangled" for graph DGMs.

**Our approach.** This observation then raises the second question: *Given a pretrained yet entangled DGM for graph data, is there a flexible framework enabling the graph controllable generation in an unsupervised manner?* To tackle this problem, we propose a model-agnostic framework coined GraphCG for unsupervised graph controllable generation. GraphCG has two main phases, as illustrated in Figure 1. During the learning phase (Figure 1(a)), GraphCG starts with the assumption that the steerable factors can be learned by maximizing the mutual information (MI) among the semantic directions. We formulate GraphCG using an energy-based model (EBM), which offers a large family of solutions. Then during the inference phase, with the learned semantic directions, we can carry out the editing task by moving along the direction with certain step sizes. As the example illustrated in Figure 1(b), the graph structure (hydroxyl group) changes consistently along the learned editing direction. For evaluation, we qualitatively verify the learned directions of five pretrained graph DGMs. Particularly for the molecular datasets, we propose a novel evaluation metric called sequence monotonic ratio (SMR) to quantitatively measure the structure change over the output sequences.

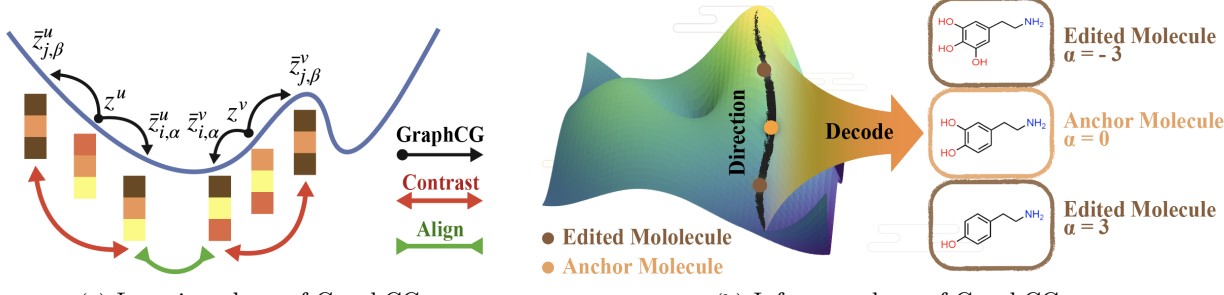

(a) Learning phase of GraphCG.  (b) Inference phase of GraphCG.

Figure 1:  (a) The learning phase. Given two latent codes $\boldsymbol{z}^u$ and $\boldsymbol{z}^v$, we edit the four latent representations along $i$-th and $j$-th direction with step size $\alpha$ and $\beta$ respectively. The goal of GraphCG, is to align the positive pair ($\bar{\boldsymbol{z}}_{i,\alpha}^u$ and $\bar{\boldsymbol{z}}_{i,\alpha}^v$), and contrast them with $\bar{\boldsymbol{z}}_{j,\beta}^u$ and $\bar{\boldsymbol{z}}_{j,\beta}^v$ respectively. (b) The inference phase. We will first sample an anchor molecule and adopt the learned directions in the learning phase for editing. With step size $\alpha \in [-3, 3]$, we can generate a sequence of molecules. Specifically, after decoding, there is a functional group change shown up: the number of hydroxyl groups decreases along the sequence in the decoded molecules.

**Our contributions.**  (1) We conduct an empirical study on the disentanglement property of three pretrained graph DGMs using six metrics, and we observe that the latent space of these pretrained graph DGMs is entangled. (2) We propose a model-agnostic method called GraphCG for the unsupervised graph controllable generation or graph editing.  GraphCG aims at learning the steerable factors by maximizing the mutual information among corresponding directions, and its outputs are sequences of edited graphs. (3) We quantitatively verify that GraphCG outperforms four competitive baselines when evaluated on two pretrained graph DGMs over two molecule datasets. (4) We further qualitatively strengthen the effectiveness of GraphCG by illustrating seven semantic factors on five pretrained graph DGMs over five graph datasets, including two for molecular graphs and three for point clouds.

**Related work.**  Recent works leverage the DGMs for various controllable generation tasks (Chen et al., 2018a; Xia et al., 2021), where the inherent assumption is that the learned latent representations encode rich semantics, and thus traversal in the latent space can help steer factors of data (Jahanian et al., 2019; Shen et al., 2020b; Härkönen et al., 2020).  Among them, one research direction (Shen et al., 2020b; Nie et al., 2021) is using supervised signals to learn the semantic-rich directions, and most works on editing the graph data focus on the supervised setting (Jin et al., 2020a; Veber et al., 2002; You et al., 2018). However, these approaches can not be applied to many realistic scenarios where extracting the supervised labels is difficult.  Another research line (Härkönen et al., 2020; Shen & Zhou, 2021; Ren et al., 2021) considers discovering the latent semantics in an unsupervised manner, but these unsupervised methods are designed to be either model-specific or task-specific, making them not directly generalizable to the graph data. A more comprehensive discussion is in Appendix B.

## 2  Background and Problem Formulation

**Graph and deep generative models (DGMs).** Each graph data (including nodes and edges) is denoted as $\boldsymbol{x} \in \mathcal{X}$, where $\mathcal{X}$ is the data space, and DGMs learn the data distribution, *i.e.*, $p(\boldsymbol{x})$. Our proposed graph editing method (GraphCG) is model-agnostic or DGM-agnostic, so we briefly introduce the mainstream DGMs for graph data as below.  Variational auto-encoder (VAE) (Kingma & Welling, 2013; Higgins et al., 2017) measures a variational lower bound of $p(\boldsymbol{x})$ by introducing a proposal distribution; flow-based model (Dinh et al., 2014; Rezende & Mohamed, 2015) constructs revertible encoding functions such that the data distribution can be deterministically mapped to a prior distribution. Note that these mainstream DGMs, either explicitly or implicitly, contain an encoder ($f(\cdot)$) and a decoder ($g(\cdot)$) parameterized by neural networks:

$$\boldsymbol{z} = f(\boldsymbol{x}), \qquad \boldsymbol{x}' = g(\boldsymbol{z}), \tag{1}$$

where $\boldsymbol{z} \in \mathcal{Z}$ is the latent representation, $\mathcal{Z}$ is the latent space, and $\boldsymbol{x}'$ is the reconstructed output graph. Since in the literature (Shen & Zhou, 2021; Shen et al., 2020b), people also call latent representations as latent

codes or latent vectors, in what follows, we will use these terms interchangeably. Note that the encoding and decoding functions in Equation (1) ($f, g$) can be stochastic depending on the DGMs we are using.

**Steerable factors.** The steerable factors are key attributes of DGMs, referring to the semantic information of data that we can explicitly discover from the pretrained DGMs. For instance, existing works (Shen & Zhou, 2021; Ren et al., 2021) have shown that using unsupervised methods on facial image DGM can discover factors such as the size of eyes, smiles, noses, etc. In this work, we focus on the steerable factors of graph data, which are data- and task-specific. Yet, there is one category of factor that is commonly shared among all the graph data: the **structure** factor. Concretely, these steerable factors can be the functional groups in molecular graphs and shapes or sizes in point clouds. The details of these steerable factors are in Appendix C.

**Semantic direction and step size.** To learn the steerable factors using deep learning tools, we will introduce the semantic directions defined on the latent space of DGM. In such a space $\mathcal{Z}$, we assume there exist $D$ semantically meaningful direction vectors, $\boldsymbol{d}_i$ with $i \in \{0, 1, \ldots, D-1\}$.[1] There is also a scalar variable, step size $\alpha$, which controls the degree to edit the sampled data with desired steerable factors (as will be introduced below), and we follow the prior work (Shen & Zhou, 2021) on taking $\alpha \in [-3, 3]$. Each direction corresponds to one or multiple factors, such that by editing the latent vector $\boldsymbol{z}$ along $\boldsymbol{d}_i$ with step size $\alpha$, the reconstructed graph will possess the desired structural modifications. The editing with a sequence of step sizes $\alpha \in [-3, 3]$ along the same direction $\boldsymbol{d}_i$ leads to a sequence of edited graphs.

**Problem formulation: graph editing or graph controllable generation.** Given a *pretrained* DGM (*i.e.*, the encoder and decoder are fixed), our goal is to learn the most semantically rich directions ($\boldsymbol{d}_i$) in the latent space $\mathcal{Z}$. Then for each latent code $\boldsymbol{z}$, with the $i$-th semantic direction and a step size $\alpha$, we can get an edited latent vector $\bar{\boldsymbol{z}}_{i,\alpha}$ and edited data $\bar{\boldsymbol{x}}'$ after decoding $\bar{\boldsymbol{z}}_{i,\alpha}$, as:

$$\boldsymbol{z} = f(\boldsymbol{x}), \qquad \bar{\boldsymbol{z}}_{i,\alpha} = h(\boldsymbol{z}, \boldsymbol{d}_i, \alpha), \qquad \bar{\boldsymbol{x}}' = g(\bar{\boldsymbol{z}}_{i,\alpha}), \tag{2}$$

where $\boldsymbol{d}_i$ and $h(\cdot)$ are the edit direction and edit functions that we want to learn. We expect that $\bar{\boldsymbol{z}}_{i,\alpha}$ can inherently possess certain steerable factors, which can be reflected in the graph structure of $\bar{\boldsymbol{x}}'$.

**Energy-based model (EBM).** EBM is a flexible framework for distribution modeling:

$$p(\boldsymbol{x}) = \frac{\exp(-E(\boldsymbol{x}))}{A} = \frac{\exp(-E(\boldsymbol{x}))}{\int_{\boldsymbol{x}} \exp(-E(\boldsymbol{x})) d\boldsymbol{x}}, \tag{3}$$

where $E(\cdot)$ is the energy function and $A$ is the partition function. In EBM, the bottleneck is the estimation of partition function $A$. It is often intractable due to the high cardinality of $\mathcal{X}$. Various methods have been proposed to handle this issue, including but not limited to contrastive divergence (Hinton, 2002), noise-contrastive estimation (Gutmann & Hyvärinen, 2010; Che et al., 2020), and score matching (Hyvärinen & Dayan, 2005; Song & Ermon, 2019; Song et al., 2020).

## 3 Entanglement of Latent Representation for Graph DGMs

In this section, we quantify the degree of disentanglement of the existing DGMs for graph data. Recall that the key intuition (Locatello et al., 2019) behind disentanglement is that a disentangled representation space should separate the distinct, informative, and steerable factors of variations in the data. In other words, each latent dimension of the disentangled representation corresponds to one or multiple factors. Therefore, the change of the disentangled dimension can lead to the consistent change in the corresponding factors of the data. This good property has become a foundational assumption in many existing controllable generation methods (Shen et al., 2020b; Shen & Zhou, 2021; Härkönen et al., 2020).

In the computer vision domain, StyleGAN (Karras et al., 2019) is one of the most recent works on image generation, and several works have proven its nice disentanglement property (Collins et al., 2020; Shen et al., 2020a; Härkönen et al., 2020; Tewari et al., 2020; Wu et al., 2021). In image generation, (Locatello et al., 2019) shows that without inductive bias, the representation learned by VAEs is not perfectly disentangled. Then the next question naturally arises: *Is the latent space of pretrained graph DGMs disentangled or not?* To answer this question, we conduct the following experiment.

---

[1]In unsupervised editing, the steerable factors on each semantic direction is known by post-training human selection.

Table 1: The six disentanglement metrics on three pretrained DGMs and two graph types. All measures range from 0 to 1, and higher scores mean more disentangled representation.

| Graph Type | DGM | Dataset | BetaVAE ↑ | FactorVAE ↑ | MIG ↑ | DCI ↑ | Modularity ↑ | SAP ↑ |
|---|---|---|---|---|---|---|---|---|
| Molecular Graph | MoFlow | ZINC250K | 0.260 | 0.175 | 0.031 | 0.953 | 0.620 | 0.009 |
| | HierVAE | ChEMBL | 0.178 | 0.165 | 0.022 | 0.114 | 0.606 | 0.026 |
| Point Cloud | PointFlow | Airplane | 0.022 | 0.025 | 0.029 | 0.160 | 0.745 | 0.022 |

There have been a series of works exploring the disentanglement of the latent space in DGMs, and here we take six widely-used ones: BetaVAE (Higgins et al., 2017), FactorVAE (Kim & Mnih, 2018), MIG (Chen et al., 2018b), DCI (Eastwood & Williams, 2018), Modularity (Ridgeway & Mozer, 2018), and SAP (Kumar et al., 2018). Each measure has its own bias, and we put a detailed comparison in Appendix C. Meanwhile, they all share the same high-level idea: given the latent representation from a pretrained DGM, they are proposed to measure how predictive it is to certain steerable factors.

To adapt them to our setting, first, we need to extract the steerable factors in graph DGMs, which requires domain knowledge. For instance, in molecular graphs, we can extract some special substructures called fragments or functional groups. These substructures can be treated as steerable factors since they are the key components of the molecules and are closely related to certain molecular properties (Seybold et al., 1987). We use RDKit (Landrum et al., 2013) to extract the 11 most distinguishable fragments as steerable factors for disentanglement measurement. For point clouds, we use PCL tool (Rusu & Cousins, 2011) to extract 75 VFH descriptors (Rusu et al., 2010) as steerable factors, which depicts the geometries and viewpoints accordingly.

Then to measure the disentanglement on graph DGMs, we consider six metrics on three datasets and two data types with three backbone models. All the metric values range from 0 to 1, and the higher the value, the more disentangled the DGM is. According to Table 1, we can observe that most of the disentanglement scores are quite low, except the DCI (Eastwood & Williams, 2018) on MoFlow. Thus, we draw the conclusion that, generally, these graph DGMs are entangled. More details of this experiment (the steerable factors on two data types and six disentanglement metrics) can be found in Appendix C.

## 4  Our Method

The analysis in Section 3 naturally raises the next research question: *Given an entangled DGM, is there a flexible way to do the graph data editing in an unsupervised manner?* The answer is positive. We propose GraphCG, a flexible model-agnostic framework to learn the semantic directions in an unsupervised manner. It starts with the assumption that the latent representations edited with the same semantic direction and step size should possess similar information (with respect to the factors) to a certain degree, thus by maximizing the mutual information among them, we can learn the most semantic-rich directions. Then we formulate this editing task as a density estimation problem with the energy-based model (EBM). As introduced in Section 2, EBM covers a broad range of solutions, and we further propose GraphCG-NCE by adopting the noise-contrastive estimation (NCE) solution.

### 4.1  GraphCG with Mutual Information

**Motivation: learning semantic directions using MI on entangled DGM.** Recall that our ultimate goal is to enable graph editing based on semantic vectors. Existing deep generative models are entangled, thus obtaining such semantic vectors is a nontrivial task. To handle this problem, we propose using mutual information (MI) to learn the semantics. MI measures the non-linear dependency between variables. Here we set the editing condition as containing both the semantic directions and step sizes. We assume that maximizing the MI between different conditions can maximize the shared information within each condition, *i.e.*, graphs moving along the same condition share more semantic information. The pipeline is as follows.

We first sample two latent codes in the latent space, $\boldsymbol{z}^u$ and $\boldsymbol{z}^v$. Such two latent codes will be treated as positive pairs, and their construction will be introduced in Section 4.3. Then we pick up the $i$-th semantic direction and one step size $\alpha$ to obtain the edited latent codes in the latent space $\mathcal{Z}$ as:

$$\bar{z}_{i,\alpha}^u = h(\boldsymbol{z}^u, \boldsymbol{d}_i, \alpha), \qquad \bar{z}_{i,\alpha}^v = h(\boldsymbol{z}^v, \boldsymbol{d}_i, \alpha). \tag{4}$$

Under our assumption, we expect that these two edited latent codes share certain information with respect to the steerable factors. Thus, we want to maximize the MI between $\bar{z}_{i,\alpha}^u$ and $\bar{z}_{i,\alpha}^v$. Since the MI is intractable to compute, we adopt the EBM lower bound from (Liu et al., 2022) as:

$$\mathcal{L}_{\text{MI}}(\bar{z}_{i,\alpha}^u, \bar{z}_{i,\alpha}^v) = \frac{1}{2}\mathbb{E}\Big[\log p(\bar{z}_{i,\alpha}^u|\bar{z}_{i,\alpha}^v) + \log p(\bar{z}_{i,\alpha}^v|\bar{z}_{i,\alpha}^u)\Big]. \tag{5}$$

The detailed derivation is in Appendix D. Till this step, we have transformed the graph data editing task into the estimation of two conditional log-likelihoods.

## 4.2 GraphCG with Energy-Based Model

Following Equation (5), maximizing the MI between $I\big(\bar{z}_{i,\alpha}^u; \bar{z}_{i,\alpha}^v\big)$ is equivalent to estimating the summation of two conditional log-likelihoods. We then model them using two conditional EBMs. Because these two views are in the mirroring direction, we may as well take one for illustration. For example, for the first conditional log-likelihood, we can model it with EBM as:

$$p(\bar{z}_{i,\alpha}^u|\bar{z}_{i,\alpha}^v) = \frac{\exp(-E(\bar{z}_{i,\alpha}^u, \bar{z}_{i,\alpha}^v))}{\int \exp(-E(\bar{z}_{i,\alpha}^{u'}, \bar{z}_{i,\alpha}^v))d\bar{z}_{i,\alpha}^{u'}} = \frac{\exp(f(\bar{z}_{i,\alpha}^u, \bar{z}_{i,\alpha}^v))}{A_{ij}}, \tag{6}$$

where $E(\cdot)$ is the energy function, $A_{ij}$ is the intractable partition function, and $f(\cdot)$ is the negative energy. The energy function is flexible and we use the dot-product:

$$f(\bar{z}_{i,\alpha}^u, \bar{z}_{i,\alpha}^v) = \langle h(z^u, d_i, \alpha), h(z^v, d_i, \alpha)\rangle, \tag{7}$$

where $h(\cdot)$ is the editing function introduced in Equation (2). Similarly for the other conditional log-likelihood term, and the objective becomes:

$$\mathcal{L}_{\text{GraphCG}} = \mathbb{E}\Big[\log \frac{\exp(f(\bar{z}_{i,\alpha}^u, \bar{z}_{i,\alpha}^v))}{A_{ij}} + \log \frac{\exp(f(\bar{z}_{i,\alpha}^v, \bar{z}_{i,\alpha}^u))}{A_{ji}}\Big]. \tag{8}$$

With Equation (8), we are able to learn the semantically meaningful direction vectors. We name this unsupervised graph controllable generation framework as GraphCG. In specific, GraphCG utilizes EBM for estimation, which yields a wide family of solutions, as introduced below.

## 4.3 GraphCG with Noise Contrastive Estimation

We solve Equation (8) using the noise contrastive estimation (NCE) (Gutmann & Hyvärinen, 2010). The high-level idea of NCE is to transform the density estimation problem into a binary classification problem that distinguishes if the data comes from the introduced noise distribution or from the true distribution. NCE has been widely explored for solving EBM (Song & Kingma, 2021), and we adopt it as GraphCG-NCE by optimizing the following objective function:

$$\begin{aligned}\mathcal{L}_{\text{GraphCG-NCE}} = -\Big(&\mathbb{E}_{p_n(\bar{z}_{j,\beta}^u|\bar{z}_{i,\alpha}^v)}\big[\log\big(1 - \sigma(f(\bar{z}_{j,\beta}^u, \bar{z}_{i,\alpha}^v))\big)\big] + \mathbb{E}_{p_{\text{data}}(\bar{z}_{i,\alpha}^u|\bar{z}_{i,\alpha}^v)}[\log\sigma(f(\bar{z}_{i,\alpha}^u, \bar{z}_{i,\alpha}^v))]\\ +&\mathbb{E}_{p_n(\bar{z}_{j,\beta}^v|\bar{z}_{i,\alpha}^u)}\big[\log\big(1 - \sigma(f(\bar{z}_{j,\beta}^v, \bar{z}_{i,\alpha}^u))\big)\big] + \mathbb{E}_{p_{\text{data}}(\bar{z}_{i,\alpha}^v|\bar{z}_{i,\alpha}^u)}[\log\sigma(f(\bar{z}_{i,\alpha}^v, \bar{z}_{i,\alpha}^u))]\Big),\end{aligned} \tag{9}$$

where $p_{\text{data}}$ is the empirical data distribution and $p_n$ is the noise distribution (derivations are in Appendix D). Recall that the latent code pairs $(z^u, z^v)$ are given in advance, and the noise distribution is on the semantic directions and step sizes. In specific, the step sizes ($\alpha \neq \beta$) are randomly sampled from [-3, 3], and the latent direction indices ($i \neq j$) are randomly sampled from $\{0, 1, ..., D\text{-}1\}$. Equation (9) is for one latent code pair, and we take the expectation of it over all the pairs sampled from the dataset. Besides, we would like to consider extra similarity and sparsity constraints as:

$$\mathcal{L}_{\text{sim}} = \mathbb{E}_{i,j}[\text{sim}(d_i, d_j)], \qquad \mathcal{L}_{\text{sparsity}} = \mathbb{E}_i[\|d_i\|], \tag{10}$$

where $\text{sim}(\cdot)$ is the similarity function between two latent directions, and we use the dot product. By minimizing these two regularization terms, we can make the learned semantic directions more diverse and sparse. Putting them together, the final objective function is:

$$\mathcal{L} = c_1 \cdot \mathbb{E}_{u,v}[\mathcal{L}_{\text{GraphCG-NCE}}] + c_2 \cdot \mathcal{L}_{\text{sim}} + c_3 \cdot \mathcal{L}_{\text{sparsity}}, \tag{11}$$

where $c_1, c_2, c_3$ are coefficients, and we treat them as three hyperparameters (check Appendix E). The above pipeline is illustrated in Figure 1, and for the next we will discuss certain key modules.

**Latent code pairs, positive and negative views.** We consider two options for obtaining the latent pairs. (1) *Perturbation (GraphCG-P)* is that for each data point $\boldsymbol{x}$, we obtain its latent code $\boldsymbol{z} = f(\boldsymbol{x})$. Then we apply two perturbations (*e.g.*, adding Gaussian noise) on $\boldsymbol{z}$ to get two perturbed latent codes as $\boldsymbol{z}^u$ and $\boldsymbol{z}^v$, respectively. (2) *Random sampling (GraphCG-R)* is that we encode two randomly sampled data points from the empirical data distribution as $\boldsymbol{z}^u$ and $\boldsymbol{z}^v$ respectively. Perturbation is one of the widely-used strategies (Karras et al., 2019) for data augmentation, and random sampling has been widely used in the NCE (Song & Kingma, 2021) literature. Then we can define the positive and negative pairs in GraphCG-NCE, where the goal is to align the positives and contrast the negatives. As described in Equation (9), the positive pairs are latent pairs moving with the same semantic direction and step size, while the negative pairs are the edited latent codes with different semantic directions and/or step sizes.

**Semantic direction modeling.** We first randomly draw a *basis vector* $\boldsymbol{e}_i$, and then model the semantic direction $\boldsymbol{d}_i$ as $\boldsymbol{d}_i = \mathrm{MLP}(\boldsymbol{e}_i)$, where $\mathrm{MLP}(\cdot)$ is the multi-layer perceptron network.

**Design of editing function.** Given the semantic direction and two views, the next task is to design the editing function $h(\cdot)$ in Equation (2). Since our proposed GraphCG is flexible, and the editing function determines the energy function Equation (7), we consider both the linear and non-linear editing functions as:

$$\bar{z}_i = \boldsymbol{z} + \alpha \cdot \boldsymbol{d}_i, \qquad \bar{z}_i = \boldsymbol{z} + \alpha \cdot \boldsymbol{d}_i + \mathrm{MLP}(\boldsymbol{z} \oplus \boldsymbol{d}_i \oplus [\alpha]), \tag{12}$$

where $\oplus$ is the concatenation of two vectors. Noticing that for the non-linear case, we are adding an extra term by mapping from the latent code, semantic direction, and step-size simultaneously. We expect that this could bring in more modeling expressiveness in the editing function. For more details, *e.g.*, the ablation study to check the effect on the design of the views and editing functions, please refer to Appendices F and G, while more potential explorations are left for future work.

## 4.4 Implementations Details

During training, the goal of GraphCG is to learn semantically meaningful direction vectors together with an editing function in the latent space, as in Algorithm 1. Then we need to manually annotate the semantic directions concerning the corresponding factors using certain post-training evaluation metrics. Finally, for the inference phase, provided with the pretrained graph DGM and a selected semantic direction (together with a step size) learned by GraphCG, we can sample a graph -> conduction editing in the latent space -> decoding to generate the edited graph, as described in Equation (2). The detailed algorithm is illustrated in Algorithm 2. Next, we highlight several key concepts in GraphCG and briefly discuss the differences from other related concepts.

**NCE and contrastive representation learning.** GraphCG-NCE is applying EBM-NCE, which is essentially a contrastive learning method, and another dominant contrastive loss is the InfoNCE (Oord et al., 2018). We summarize their relations below. (1) Both contrastive methods are doing the same thing: align the positive pairs and contrast the negative pairs. (2) EBM-NCE (Hassani & Khasahmadi, 2020; Liu et al., 2022) has been found to outperform InfoNCE on certain graph applications like representation learning. (3) What we want to propose here is a flexible framework. Specifically, EBM pro-

---

**Algorithm 1** Learning Phase of GraphCG

1: **Input:** Given a pretrained generative model encoder, $f(\cdot)$.
2: **Output:** Learned direction vector $\boldsymbol{d}_i$ and function $h(\cdot)$.
3: Select latent codes $\boldsymbol{z}^u, \boldsymbol{z}^v \in \mathcal{Z}$ from empirical dataset and $f(\cdot)$.
4: **for** each step size $\alpha$ and each direction $i$ **do**
5:     Set $\bar{z}_{i,\alpha}^u = h(\boldsymbol{z}^u, \boldsymbol{d}_i, \alpha)$.
6:     Set $\bar{z}_{i,\alpha}^v = h(\boldsymbol{z}^v, \boldsymbol{d}_i, \alpha)$.
7:     Assign positive to pair $(\bar{z}_{i,\alpha}^u, \bar{z}_{i,\alpha}^v)$.
8:     **for** step size $\beta \neq \alpha$ and direction $j \neq i$ **do**
9:         Set $\bar{z}_{j,\beta}^u = h(\boldsymbol{z}^u, \boldsymbol{d}_j, \beta)$.
10:        Set $\bar{z}_{j,\beta}^v = h(\boldsymbol{z}^v, \boldsymbol{d}_j, \beta)$.
11:        Assign negative to pair $(\bar{z}_{i,\alpha}^u, \bar{z}_{j,\beta}^v)$.
12:        Assign negative to pair $(\bar{z}_{j,\beta}^u, \bar{z}_{i,\alpha}^v)$.
13:     **end for**
14:     Do SGD w.r.t. GraphCG in Equation (11).
15: **end for**

---

vides a more general framework by designing the energy functions, and EBM-NCE is just one effective so-

lution. Other promising directions include the denoising score matching or denoising diffusion model (Song et al., 2020), while InfoNCE lacks such a nice extensibility attribute.

**GraphCG and contrastive self-supervised learning (SSL).** GraphCG shares certain similarities with the self-supervised learning (SSL) method, however, there are some inherent differences, as summarized below. (1) SSL aims at learning the data representation by operating data augmentation on the data space, such as node addition and edge deletion. GraphCG aims at learning the semantically meaningful directions by editing on the latent space (the representation function is pretrained and fixed).

---

**Algorithm 2** Inference Phase of GraphCG

1: **Input:** Given a pre-trained generative model, $f(\cdot)$ and $g(\cdot)$, a learned direction vector $\boldsymbol{d}$.
2: **Output:** A sequence of edited graphs.
3: Sample $\boldsymbol{z}$ with DGM or $\boldsymbol{x}$ from a graph dataset.
4: If the latter, get a latent code $\boldsymbol{z} = f(x)$.
5: **for** step size $\alpha \in [-3, 3]$ **do**
6:    Do graph edit in the latent space to get $\bar{\boldsymbol{z}}_{i,\alpha} = h(\boldsymbol{z}, \boldsymbol{d}, \alpha)$.
7:    Decode to the graph space with $\bar{\boldsymbol{x}}' = g(\bar{\boldsymbol{z}}_{i,\alpha})$.
8: **end for**
9: Output is thus a sequence of edited graphs, $\{\bar{\boldsymbol{x}}'\}$.

---

(2) Based on the first point, SSL aims at using different data points as the negative samples. GraphCG, on the other hand, is using different directions and step-sizes as negatives. Namely, SSL is learning data representation in the inter-data level, and GraphCG is learning the semantic directions in the inter-direction level.

**Output sequence in the discrete space.** Recall that during inference time (Algorithm 2), GraphCG takes a DGM and the learned semantic direction to output a sequence of edited graphs. Compared to the vision domain, where certain models (Shen & Zhou, 2021; Shen et al., 2020b) have proven their effectiveness in many tasks, the backbone models in the graph domain have limited discussions. This is challenging because the graph data is in a discrete and structured space, and the evaluation of such space is non-trivial. Meanwhile, GraphCG essentially provides another way to verify the quality of graph DGMs. GraphCG paves the way for this potential direction, and we would like to leave this for future exploration.

## 5 Experiments

In this section, we show both the qualitative and quantitative results of GraphCG, on two types of graph data: molecular graphs and point clouds. Due to the page limit, We put the experiment and implementation details in Appendix E.

### 5.1 Graph Data: Molecular Graphs

**Background of molecular graphs.** A molecule can be naturally treated as a graph, where the atoms and bonds are nodes and edges, respectively. The unsupervised graph editing tasks can thus be formulated as editing the substructures of molecular graphs. In practice, people are interested in substructures that are critical components of molecules, which are called the 'fragments'. In recent years, graph representation

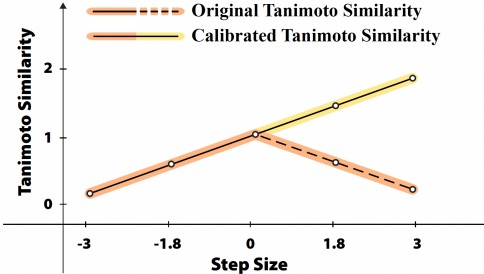

$$\phi_{\mathrm{SMR}}\left(\{s(\bar{\boldsymbol{x}}')\}_i^m, \gamma, \tau\right) = \begin{cases} 1, & \mathrm{len}\Big(\mathrm{set}\big(\{s(\bar{\boldsymbol{x}}')\}_i^m\big)\Big) \geq \gamma \\ & \quad \wedge \mathrm{monotonic}_{\tau}\big(\{s(\bar{\boldsymbol{x}}')\}_i^m\big), \\ 0, & \mathrm{otherwise} \end{cases} \tag{13}$$

$$\phi_{\mathrm{SMR}}(\gamma, \tau)_i = \frac{1}{M} \sum_{m=1}^{M} \phi_{\mathrm{SMR}}\left(\{s(\bar{\boldsymbol{x}}')\}_i^m, \gamma, \tau\right), \tag{14}$$

$$\text{top-K}(\gamma, \tau) = \frac{1}{K} \sum_{i \in \text{top-K directions}} \left(\phi_{\mathrm{SMR}}(\gamma, \tau)_i\right). \tag{15}$$

Figure 2: This illustrates the sequence monotonic ratio (SMR) on calibrated Tanimoto similarity (CTS). Equations (13) and (14) are the SMR on each sequence and each direction respectively, where $M$ is the number of generated sequences for the $i$-th direction and $\{s(\bar{\boldsymbol{x}}')\}_i^m$ is the CTS of the $m$-th generated sequence with the $i$-th direction. Equation (15) is the average of top-K SMR on $D$ directions.

learning has been extensively explored on the molecular graph (Duvenaud et al., 2015; Gilmer et al., 2017; Liu et al., 2018; Yang et al., 2019b; Corso et al., 2020).

**Backbone DGMs.** We consider two state-of-the-art DGMs for molecular graph generation. MoFlow (Zang & Wang, 2020) is a flow-based generative model on molecules that adopts an invertible mapping between the input molecular graphs and a latent prior. HierVAE (Jin et al., 2020a) is a hierarchical VAE model that encodes and decodes molecule atoms and motifs in a hierarchical manner. Besides, the pretrained checkpoints are also provided, on ZINC250K (Irwin & Shoichet, 2005) and ChEMBL (Mendez et al., 2019), respectively.

**Editing sequences and anchor molecule.** As discussed in Section 4, the output of the inference in GraphCG, is a sequence of edited molecules with the $i$-th semantic direction, $\{\bar{\boldsymbol{x}}'\}_i$. We first randomly generate a molecule using the backbone DGMs (without the editing operation), and we name such molecule as the anchor molecule, $\bar{\boldsymbol{x}}^*$. Then we take 21 step sizes from -3 to 3, with interval 0.3, to obtain a sequence of 21 molecules following Equation (2). Note that the edited molecule with step size 0 under the linear editing function is the same as the anchor molecule, $i.e.$, $\bar{\boldsymbol{x}}^*$.

**Change of structure factors and evaluation metrics.** We are interested in the change of the graph structure (the steerable factors) along the output sequence edited with the $i$-th semantic direction. To evaluate the structure change, we apply the Tanimoto similarity between each output molecule and the anchor molecule. Besides, for the ease of evaluating the monotonicity, we apply a Tanimoto similarity transformation on the output molecules with positive step sizes by taking the deduction from 2. We call this calibrated Tanimoto similarity (CTS) sequence, marked as $\{s(\bar{\boldsymbol{x}}')\}_i$. An illustration is shown in Figure 2. Further, we propose a metric called Sequence Monotonic Ratio (SMR), $\phi_{\text{SMR}}(\gamma, \tau)_i$, which measures the monotonic ratio of $M$ generated sequences edited with the $i$-th direction. It has two arguments: the diversity threshold $\gamma$ constrains the minimum number of distinct molecules, and the tolerance threshold $\tau$ controls the non-monotonic tolerance ratio along each sequence.

**Evaluating the diversity of semantic directions.** SMR can evaluate the monotonic ratio of output sequences generated by one direction. To better illustrate that GraphCG is able to learn multiple directions with diverse semantic information, we also consider taking the average of top-$K$ SMR to reveal that all the best $K$ directions are semantically meaningful, as in Equation (15).

**Baselines.** For baselines, we consider four unsupervised editing methods. (1) The first is Random. It randomly samples a normalized random vector in the latent space as the semantic direction. (2) The second one is Variance. We analyze the variance on each dimension of the latent space, and select the highest one with one-hot encoding as the semantic direction. (3) The third one is SeFa (Shen & Zhou, 2021). It first decomposes the latent space into lower dimensions using PCA, and then takes the most principal components (eigenvectors) as the semantic-rich direction vectors. (4) The last one is DisCo (Ren et al., 2021). It maps each latent code back to the data space, followed by an encoder for contrastive learning, so it requires the backbone DGMs to be end-to-end and is infeasible for HierVAE.

**Quantitative results.** We take $D = 10$ to train GraphCG, and the optimal results on 100 sampled sequences are reported in Table 2. We can observe that GraphCG, can show consistently better structure change with both top-1 and top-3 directions. This can empirically prove the effectiveness of our proposed GraphCG. More comprehensive results are in Appendix F.

**Analysis on steerable factors in molecules: functional group change.** For visualization, we sample 8 molecular graph sequences along 4 selected directions in Figure 3, and the backbone DGM is HierVAE pretrained on ChEMBL. The CTS holds a good monotonic trend, and each direction shows certain unique changes in the molecular structures, $i.e.$, the steerable factors in molecules. Some structural changes are reflected in molecular properties. We expand all the details below. In Figures 3(a) and 3(b), the number of halogen atoms and hydroxyl groups (in alcohols and phenols) in the molecules decrease from left to right, respectively. In Figure 3(c), the number of amides in the molecules increases along the path. Because amides are polar functional groups, the topological polar surface area (tPSA) of the molecules also increases, which is a key molecular property for the prediction of drug transport properties, $e.g.$, permeability (Ertl et al., 2000). In Figure 3(d), the flexible chain length, marked by the number of ethylene ($CH_2CH_2$) units, increases

Table 2: This table lists the sequence monotonic ratio (SMR, %) on calibrated Tanimoto similarity (CTS) for the top-1 and top-3 directions. The best performances are marked in **bold**.

| Model | Dataset | diversity $\gamma$ | Tanimoto top-1 | | | | Tanimoto top-3 | | | |
|---|---|---|---|---|---|---|---|---|---|---|
| | | | 3 | | 4 | | 3 | | 4 | |
| | | tolerance $\tau$ | 0 | 0.2 | 0 | 0.2 | 0 | 0.2 | 0 | 0.2 |
| MoFlow | ZINC250k | Random | 23.0 | 25.0 | 12.0 | 15.0 | 22.0 | 24.0 | 11.0 | 13.7 |
| | | Variance | 24.0 | 28.0 | 12.0 | 16.0 | 20.0 | 25.0 | 10.0 | 15.0 |
| | | SeFa Shen & Zhou (2021) | 4.0 | 4.0 | 0.0 | 0.0 | 3.3 | 3.3 | 0.0 | 0.0 |
| | | DisCo Ren et al. (2021) | 7.0 | 14.0 | 2.0 | 8.0 | 5.3 | 11.7 | 2.0 | 7.7 |
| | | GraphCG-P | **32.0** | **34.0** | **16.0** | **18.0** | **29.0** | **31.0** | **13.7** | **16.3** |
| | | GraphCG-R | 25.0 | 26.0 | 11.0 | 14.0 | 22.0 | 24.3 | 10.3 | 13.3 |
| HierVAE | ChEMBL | Random | 14.0 | 45.0 | 14.0 | 43.0 | 10.0 | 42.3 | 9.3 | 41.7 |
| | | Variance | 23.0 | 59.0 | 19.0 | 55.0 | 18.3 | 52.7 | 15.7 | 50.3 |
| | | SeFa Shen & Zhou (2021) | 4.0 | 41.0 | 4.0 | 41.0 | 2.3 | 36.0 | 2.3 | 36.0 |
| | | GraphCG-P | 40.0 | **73.0** | **32.0** | **65.0** | 36.0 | **64.3** | 26.3 | **57.7** |
| | | GraphCG-R | **42.0** | 67.0 | 30.0 | 55.0 | **38.0** | 62.3 | **28.7** | 53.7 |

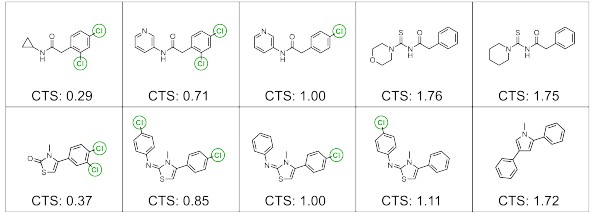

(a) Steerable factor: number of halogens.

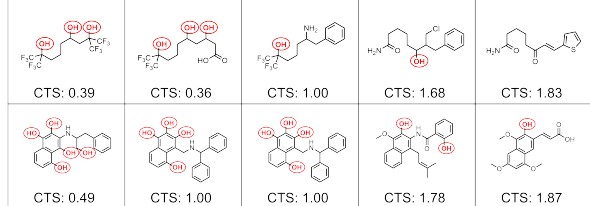

(b) Steerable factor: number of hydroxyls.

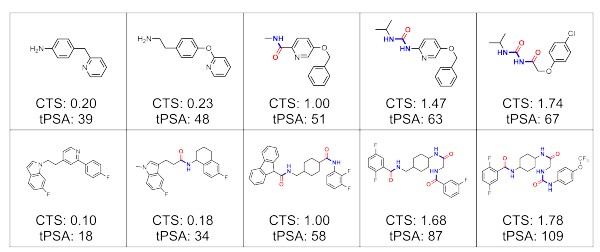

(c) Steerable factor: number of amides.

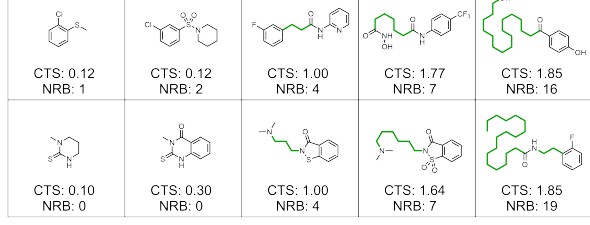

(d) Steerable factor: chain length.

Figure 3: GraphCG for molecular graph editing. We visualize the output molecules and CTS in four directions with two sequences each, where each sequence consists of five steps. The five steps correspond to five step sizes: -3, -1.8, 0, 1.8, and 3, where 0 marks the anchor molecule (center point of reach sequence). Figure 3(a) visualizes how the number of halogens (marked in green) decreses from -3 to 3. Figure 3(b) visualizes how the number of hydroxyls (marked in red) decreases from -3 to 3. Figure 3(c) visualizes how the number of amides (marked in red and blue) increases from -3 to 3. Figure 3(d) visualizes how the number of chains (marked in green) increases from -3 to 3. Notably, certain properties change with molecular structures accordingly, like topological polar surface area (tPSA) and the number of rotatable bonds (NRB).

from left to right. Since the number of rotatable bonds (NRB) measures the molecular flexibility, it also increases accordingly (Veber et al., 2002).

## 5.2 Graph Data: Point Clouds

**Background of point clouds.** A point cloud is represented as a set of points, where each point $P_i$ is described by a vector of 3D Euclidean coordinates possibly with extra channels (*e.g.*, colors, surface normals, and returned laser intensity). In 2017, Qi et. al (Qi et al., 2017) designed a deep learning framework called PointNet that directly consumes unordered point sets as inputs and can be used for various tasks such as classification and segmentation.

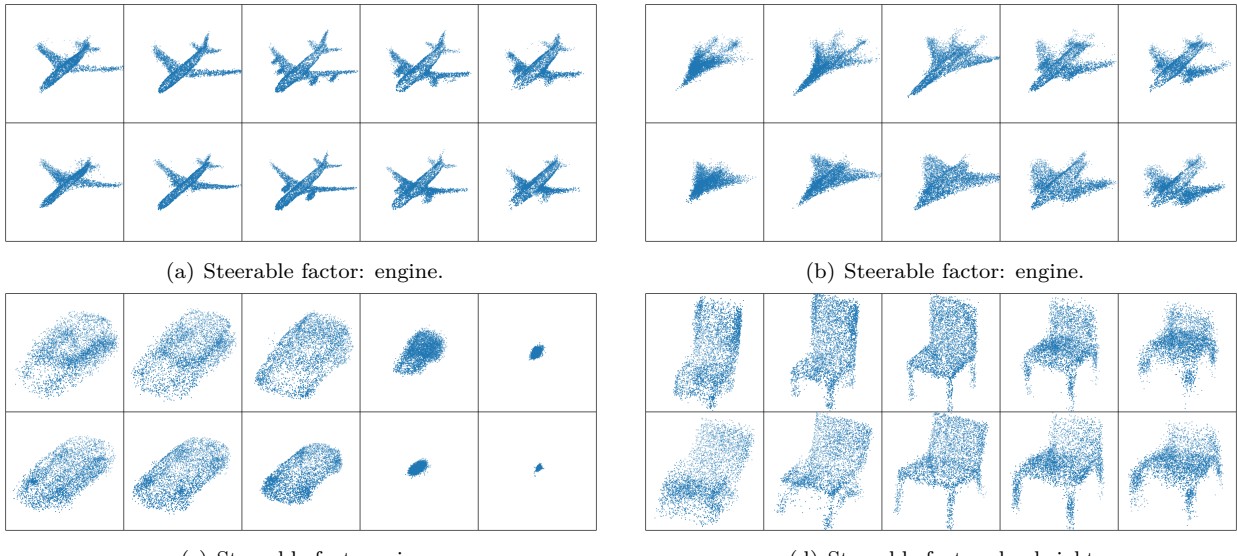

(a) Steerable factor: engine.         (b) Steerable factor: engine.

(c) Steerable factor: size.         (d) Steerable factor: leg height.

Figure 4: GraphCG for point clouds editing. We show four editing sequences, where each sequence consists of five point clouds, and the center one is the anchor point clouds, *i.e.*, with step size 0. The other four point clouds correspond to step size with -3, -1.8, 1.8, and 3, respectively. Figure 4(a) and Figure 4(b) refer the same semantic direction, and they are showing how to steer the factor engine: the number of engines will be decreased and increased with the negative (left) and positive (right) step size respectively. Similarly, Figures 4(c) and 4(d) illustrate the effect of the steerable factors on the car size and the chair leg height.

**Backbone DGMs.** We consider one of the latest DGMs on point clouds, PointFlow (Yang et al., 2019a). It is using the normalizing flow model for estimating the 3D point cloud distribution. Then we consider PointFlow pretrained on three datasets in ShapeNet (Chang et al., 2015): Airplane, Car, and Chair. All point clouds are obtained by sampling points uniformly from the mesh surface.

**Analysis on steerable factors in point clouds: shape change.** To train GraphCG, we take $D = 10$ directions, and we sample 8 point cloud sequences along 3 directions for visualization in Figure 4. More results are in Appendix G. It is observed that GraphCG, can steer the shape of the point clouds, *e.g.*, the size of cars and the height of chair legs. We also find it interesting that GraphCG, can steer more finger-trained factors, like modifying the number of jet engines.

## 6 Conclusion and Discussion

In this work, we are interested in unsupervised graph editing. It is a well-motivated task for various real-world applications, and we discuss two mainstream data types: molecular graphs and point clouds. We start with exploring the latent space of mainstream deep generative models and propose GraphCG, a model-agnostic unsupervised method for graph data editing. The key component of GraphCG, is EBM, and we take the GraphCG-NCE as the solution for now. For future work, we may as well extend it to more advanced solutions like denoising diffusion model (Ho et al., 2020).

One limitation of GraphCG, (as well as the solutions to general unsupervised data editing) (Härkönen et al., 2020; Shen & Zhou, 2021; Ren et al., 2021) is that we may need some post-training human selection (as shown in Algorithm 2) to select the most promising semantic vectors to steer factors. Another issue is the lack of open-sourced evaluation metrics. This requires both a deep understanding of the representation space of deep generative models and domain knowledge of the problem. For instance, activity cliff is a challenging task (Hu & Jurgen, 2012) for editing, while current measures fail to capture it. To set up constructive evaluation metrics can help augment our understandings from both the domain and technique perspectives. This is beyond the scope of our work, yet would be interesting to explore as a future direction.

## Code and Data Availability

The codes and data download scripts are available at this GitHub repository.

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
