# OpenReview forum: "Unsupervised Discovery of Steerable Factors When Graph Deep Generative Models Are Entangled"
_TMLR — Accepted by TMLR_

### Review · Reviewer_dKVp · 2023-09-17

**Summary Of Contributions:**

The paper has two main contributions:
- It shows that the latent representation of Graph Deep Generative Models is not disentangled/does not isolate steerable factors.
- It proposes a model agnostic method to find latent disentangled/steerable directions ex post, after the model has been trained.

**Audience:**

Yes

**Claims And Evidence:**

No

**Requested Changes:**

From most important to least important:
- __Method section (critical):__ The method section needs to be thoroughly clarified. Section 4.1. lacks necessary details to be fully understandable. Typically, the authors mention that they need to sample latent codes from the latent space. This sampling procedure will differ from model to model, but this can be done either in a data dependent fashion or in a data independent fashion. For instance, for a simple non hierarchical gaussian VAE, or a gaussian based flow based model, this sampling can be done either by sampling from the gaussian prior, or by sampling data from the dataset, then encoding those points using the encoding function. When using the first sampling method, samples do not depend on the original data distribution, or on the generative model at play, and the quantities that are mentioned below (i.e. the MI between the considered distribution) could be analytically expressed and optimized. However, this would lead to discovery of steerable factors that are independent of the model and the data distribution, which I don't think is a reasonable thing to look for. By not specifying out of the box what sampling method is considered, the method section is very hard to read and get intuition about. This confusion is made even clearer in Algorithm 1, where f and g are defined as inputs to the algorithm, but never actually used. Besides I am not sure I understand fully why optimizing mutual information to get steerable directions is a meaningful thing to do, since what we seemingly expect from them is for the directions to be, on the one hand, informative, but mostly independent from one another. Maximizing mutual information seems to only maximize informativity, but not independence. To palliate for this issue, the authors seem to introduce an additional orthogonality constraint as an afterthought.
- __Semantic direction modeling (critical):__ I don't understand why the authors are optimizing the parameters of an MLP taking as input (fixed) basis vectors, and not directly the coordinates of the d_i's. Could you clarify?
- __Experimental evaluation (critical):__  As I am not very familiar with chemistry, I cannot evaluate the qualitative evaluation on the chemistry dataset. However, on the cloud point datasets, I am a bit doubtful of the independence of the factors discovered: for all of the examples given, the form of the generated object significantly varies along with the supposed steerable factor.
- __Entanglement/Steerable factors (strengthen):__ A proper explanation of what steerable is, how it is defined, how it is measured and how it differ from disentanglement could be pushed at the beginning of the paper. I still don't understand the distinction between steerable factors and disentangled representation. If there are none, I would suggest framing the paper in term of disentanglement. I only got a clear idea of how one could define and measure steer-ability in the 'Entanglement of latent representation', which I find to be a bit late.
- __Additional experiments in other modalities (strengthen):__ Given that the method seems quite general and not restricted to the graph modality, I think adding experiments on other modalities, potentially where more comparisons would be available, would add a lot to the paper

**Strengths And Weaknesses:**

__Strengths:__
- Most sections, apart from the method section read well, present the problem at hand relatively  clearly, and succeed in explaining why this is an interesting problem.

__Weaknesses:__
- The main weakness comes from the method section, which is unclear enough to leave doubts about the validity of the method.
- Some of the experimental results are not convincing enough.

---

> ### Author Response · Authors · 2023-10-04
> **Reply (1/2)**
>
> Thank you for your thorough review of our paper. We have carefully addressed the concerns you raised, and you can find the corresponding responses below.
>
> **Method section**
>
> Thank you for raising such detailed questions. We are happy to explain in more details.
>
> - First, we are doing this in a data-dependent way, as in Eq 2. In Eq 2, you can tell that $z$ is not randomly sampled from the prior distribution, but the encoded representation of $x$, which is randomly sampled from the empirical data distribution. More concretely, we have two options for obtaining pairs of $z$ from $x$ (in Sec 4.3):
>     - Option 1 Perturbation: For each data point $x$, we obtain its latent code $z$, and do the random perturbation to get two latent codes. We didn't explain this in the first version, and will update it once later.
>     - Option 2 Random sampling: We randomly sample two data points $x$, and then obtain the corresponding latent codes $z$. This has already been explained in the first version.
>     - We will also update Algorithm 1 accordingly.
> - Second, about why we are using the mutual information (MI):
>     - You are right that MI measures the nonlinear dependency ([reference in wiki](https://en.wikipedia.org/wiki/Mutual_information)), or what you called the `informative`. [This figure](https://en.wikipedia.org/wiki/Mutual_information#/media/File:Entropy-mutual-information-relative-entropy-relation-diagram.svg) presents a nice and intuitive explanation.
>     - To adapt it to our setting, as shown in Eq 4, we are maximizing the MI between two latent codes, $z^u$ and $z^v$. These two latent codes are from one/two independent data points, and we provide two options to obtain them (this is related to the first question above).
>     - To put this in another way, this is essentially the core of GraphCG: learning the semantic directions by forcing independent latent codes to move towards the semantically same directions.
>     - Further, maximizing the MI is the conjecture in our work. And empirically, we use both the quantitative and qualitative results to support this conjecture.
>
>
> **Semantic direction modeling**
>
> Thank you for raising this question. Now, we present it as an architecture design, yet we can explain the underlying intuitions from two aspects.
>
> - From the statistics aspect, this is essentially the reparameterization trick ([reference 1](https://arxiv.org/abs/1312.6114), [reference2](https://medium.com/@llionj/the-reparameterization-trick-4ff30fe92954)).
> - From the optimization aspect. If we directly model $d_i \in \mathbb{R}^l$, where $l$ is the latent dimension, it turns out that the optimization (of neural network) is very hard, and the model can collapse easily (e.g., the output semantic directions have no meanings). Yet, once we model $d_i = \text{MLP}(e_i)$, the optimization is more stable.
>
>
>
>
> **Experimental evaluation**
>
> First, we would like to highlight that our evaluation pipeline is:
> - We train 10 directions (in learning).
> - Then, we draw the edited graphs (in inference).
> - These two steps are **unsupervised**.
> - Finally, we will manually summarize what is the semantic factor for each direction.
> - Thus, if you think what we summarize is not appropriate in Figure 4, we are happy to modify it. Currently, the steerable factors we have summarized in Figure 4 are:
>     - steerable factor engine for Figure 4.a \& 4.b.
>     - steerable factor size for Figure 4.c.
>     - steerable factor leg height for Figure 4.d.
>
> Additionally, for the chemistry results in Figure 3, we highlight the modified parts in different colors so as for readers to capture the change of steerable factors easily. We can guarantee that the results are valid.

---

> ### Author Response · Authors · 2023-10-04
> **Reply (2/2)**
>
> **Entanglement and Steerable Factors**
>
> Thank you for bringing up this question. In the current draft, we have elucidated the steerable factors and entanglement, as discussed below. Additionally, we will distinctly emphasize and relocate the definition of steerable factors to Section 1.
>
> - We have defined the steerable factors in Sec 2. We have also provided detailed steerable factors for each graph data in Sec C.1.
> - We have defined the entanglement in Sec 3.
> - To summarize their relationship:
>     - If the DGM is disentangled, then we can easily find the steerable factors by doing editing on each of the latent dimensions in $z$.
>     - If the DGM is entangled (Sec 3), then the editing task becomes more challenging. This motivates us to propose GraphCG (Sec 4).
>
>
>
> **Additional experiments**
>
> Thank you for posing this question. Below, we provide a detailed explanation supported by specific reasons and materials on the choice of (not) including image editing in this paper.
>
> 1. The GraphCG method is very general and can be applied to other applications like image editing.
> 2. But, we want to point out that, for the image generation model like StyleGAN, the model itself is perfectly disentangled. While in this paper, we are focusing on the general setting, when the graph deep generative models are entangled (as in the title).
> 3. Thus, though GraphCG can be applied to other applications like image editing, it is beyond the scope of this paper.
> 4. Meanwhile, we still provide some preliminary results on image editing: [example 1](https://anonymous.4open.science/r/GraphCG_outputs-075A/image_age_gender.pdf), [example 2](https://anonymous.4open.science/r/GraphCG_outputs-075A/image_pose.pdf), [example 3](https://anonymous.4open.science/r/GraphCG_outputs-075A/image_smile_lip.pdf), [example 4](https://anonymous.4open.science/r/GraphCG_outputs-075A/image_tower_style.pdf).
> 5. For more details, please check `Scope of GraphCG: Why Not Image` on page 15 in the appendix. We have listed more comprehensive reasons and references for readers to check.

---

> > ### Author Response · Authors · 2023-11-13
> > **Manuscript Updated**
> >
> > Thank you for your comments again. We have now modified the manuscript (highlighted in green) to address your comments, especially on how to obtain the latent code pairs $z^u$ and $z^v$, and where the encoder $f$ is used in the algorithm. Please feel free to check, and we are delighted to address any additional questions you may have.

---

### Review · Reviewer_cmNf · 2023-10-01

**Summary Of Contributions:**

The work introduces GraphCG a framework to understand the structure of a generative model, and provide means to edit examples by moving in meaningful directions. The core idea is to use a contrastive loss, and rely on ideas around mutual information and energy based models to identify semantically meaningful directions d to move in the latent space and to learn an editing function h (which can be linear) that allows moving in this directions. Specifically the work focuses on performing this on graph data, which is of particular importance for molecule editing.

**Audience:**

Yes

**Broader Impact Concerns:**

The paper proposes a technique that allows editing graphs in a semantically meaningful way. A target application widely discussed in the paper is molecule editing. As with any technology of this form, it has a dual use. It could be used for drug discovery and provide solutions for existing illnesses, as well as possibly for creating viruses. Or more generally, such techniques can be used on social graphs to understand how to modify them to maximize certain properties.
However neither of these applications are a straight forward application of this work. And in particular, there exists a large body of works in this space, to which the current publication belongs, and this particular paper does not stands out from an ethical implications point of view compared to the others.

Therefore I think there are no ethical implications that would raise specific concerns for this work.

**Claims And Evidence:**

Yes

**Requested Changes:**

I think the paper could be accepted in its current form. I'm not sure changes are strictly required.

To strengthen the paper I think a more detailed appendix E could help (e.g. what does norm stand for? whose norm? which norm?). Maybe some snippets of code could help there. In general how you initialize these directions, ablation or discussion about robustness of hyperparameters choices.

**Strengths And Weaknesses:**

The paper is well written and provides sufficient context and is properly placed within the existing literature, to allow the reader to understand the motivation as well as the main intuitions behind the method.

To me a potential weakness is that the presentation of the paper tends to be somewhat generic. On one hand side, this makes a lot of sense, the framework introduced is generic, and not specific to a particular generative model. On the other hand, grounding things a little bit more can help understand what is going on (e.g. specifying things in terms of a particular generative model, or talk about the exact form of the function h). Some of these details are pushed towards appendix E, but even there I think things are somewhat brief. Overall I do think that someone familiar with the topic could figure out these details and reproduce most results, but someone less familiar might have a much harder time figuring out where to start. There is an algorithm provided in the paper, but I still think it keeps things quite generic.

---

> ### Author Response · Authors · 2023-10-04
> **Reply**
>
> Thank you for your support of our work!
>
> We plan to enhance the paper by incorporating more graph-specific details and relocating the related discussions from Section E to the main body.

---

### Review · Reviewer_jtQQ · 2023-11-05

**Summary Of Contributions:**

### _Summary_
This paper introduces GraphCG, a new method for unsupervised graph controllable generation that can be applied to reveal steerable factors in pretrained deep generative models (DGMs) for graph data. The main contributions lie in the identification and utilization of steerable factors within these latent spaces. They do this by maximizing mutual information among semantically-rich directions, enabling the generation of modified graphs that vary along the learned factors. The authors’ method is tested on two datasets (molecules, point clouds) and the authors further provide experiments to quantify disentanglement of the different pretrained graph models.

Overall, the paper tackles an important problem in unsupervised graph controllable generation and editing, and provides a well motivated solution to the problem. However, it requires improvements in clarity, contextualization, and justification to solidify its contributions to the field. Further comments and questions are provided below.

**Audience:**

Yes

**Broader Impact Concerns:**

No broader impact concerns are noted.

**Claims And Evidence:**

Yes

**Requested Changes:**

### _Critical requested changes_
1. Please provide the requested revisions to the background, method description and related work as requested above to improve clarity of the presentation.
2. Experiments and insights into the stability of the method in discovering the same steerable factors.
3. More discussion of related work and other approaches to controllable generation on graphs, and related work in disentangled factors for point clouds.
4. Discussion of the hyperparameters and the different components of the loss in Eq (11) and how they impact the factors and performance. Visualizations and examples would be helpful (see additional suggestions below).
5. Additional explanations of the molecular graph problem and datasets, and further insights into how to interpret your results on these datasets.

### _Additional requested changes_
1. Additional experiments on more datasets to further highlight the application of the method.
2. Quantification of the stability of steerable factors as you ablate different parts of the loss in Eq (11)

**Strengths And Weaknesses:**

### _Main comments_
1. __Related work:__  The presentation of the background and related work needs to be significantly expanded to fully understand the problem formulation and contextualize the contributions of GraphCG. Currently, there is very little discussion of other approaches for graph editing, and the discussion of the datasets and experiments is lacking, thus the interpretation of their results, especially for the molecular examples is difficult.

2. __Ablations of losses in Eq 11:__ The loss in Eq (11) has a number of components. However, it’s unclear how these different components impact the performance of their steering method.

3. __Clarity of problem formulation and method description:__ The authors' description of the problem is lacking in detailed assumptions and clear definitions, especially regarding the 'steerable factors' and their manifestation within graph structures. The formulation appears to be intuitive rather than rigorously defined, which makes it hard to understand precisely what the method entails. By the time we get to Equation (12), it seems to come too late. The reader is also left confused for some time about what a “semantic direction is”, how positive and negative views are generated, and how the directions are learned. The small subsection on “semantic direction modeling” should be integrated with the subsection w Eq (12) and not left alone without context.

4. __Assumptions behind disentanglement experiment:__  The assumption that certain molecular fragments can serve as steerable factors and the subsequent use of RDKit for their extraction lacks justification. It would be useful to have more explanation of why these fragments are chosen and how they indeed reflect the latent representations’ structure. How can you be sure that these initial preprocessing steps are correct, and what if the DGM is disentangled with respect to different sets of factors?

5. __Experimental details:__  The lack of discussion on the experimental setup and datasets within the main text makes it difficult to understand and assess the impact of the results.

6. __Accessibility for non-experts:__ Finally, the paper would benefit from a more thorough walk through of the experiments and results concerning these datasets to enhance its accessibility and ensure that readers across various domains can really appreciate the contributions. The results and experiments on molecular datasets are not easy to understand without domain expertise.

### _Additional questions:_
1. How stable are the learned factors? Please provide some information and experiments to document the consistency of the factors and the sensitivity to hyperparameters.

2. How does the number of learned semantic directions impact performance and interpretable factors? Are there cases where a model doesn’t learn a good set of representations and your approach can provide insight into these failure modes?

3. “GraphCG aims at learning the steerable factors by maximizing the mutual information among corresponding directions, and its outputs are sequences of edited graphs.” - Make more clear what you mean by “sequences of edited graphs” at this point, consider using figures of point clouds later to illustrate your point

4. How is the method connected to other applications of contrastive learning on graphs?

---

> ### Author Response · Authors · 2023-11-13
> **Reply (1/2)**
>
> Thank you for raising these questions, and we have carefully addressed them as follows.
>
> ### Related work.
>
> Thank you for raising this question.
> - We are happy to discuss other **unsupervised graph editing** papers if you can provide the detailed references.
> - The unsupervised editing has only been recently studied on the vision tasks. These works are not that highly related, thus we put them in Sec B. Feel free to check this.
> - For the lack of discussion on datasets and experiments, we have them explaiend in Sec 5, A, and C. If you can help point out which specific points that confuse you, we are happy to answer them in more details.
>
>
> ### Ablations of losses in Eq 11.
>
> Thank you for raising this question. We have the ablation study with qualitative results in Sec F.4.
>
> ### Clarity of problem formulation and method description
>
> Thank you for raising this question. We have provided most of these details in Sec 2 and appendix.
>
> - The first problem formulation is in Sec 2 (Eq 2). Eq 12 is showing the detailed expansion of Eq 2.
> - We have provided the definition of steerable factors in Sec 2 and C.
> - The steerable factors and emantic direction are indeed intuitive factors, rather than rigorous equations. We have them explained them in Sec 1, Sec 2, and Figure 1 to provide a high-level untuition for the readers.
>     - First, we need to keep in mind that for this research line, all the steerable factors are learned in an `unsupervised manner`, i.e., we need to manually detect the steerable factors only after we learn the factors. We have highlighted this in the revised version.
>     - Then additionally, the steerable factors are `model- and data-dependent`.
>         - For example in the unsupervised image editing tasks: In the facial expression editing tasks, the factors can include the size of eyes, smiles, noses, etc. In the cartoon image editing tasks, the factors can include include the character poses, the eyes, the gender, etc.
>         - Then to adopt this for the unsupervised graph editing: We have listed all the potential factors in Figure 3 & 4, and Sec C.
> - The construction of positive and negative views are in Sec 4.3, between Eq 11 and Eq 12.
> - The direction learning is in Eq 11 (the objective function).
>
>
> ### Assumptions behind disentanglement experiment
>
> This question is about why we choose using the steerable factors for molecules in Sec C.1.
>
> - Initially, we consider all the 85 fragments provided in rdkit. The link is [here](https://rdkit.org/docs/source/rdkit.Chem.Fragments.html).
> - To calculate the disentanglement for molecules, we randomly sample 10k molecules on QM9-MolFlow, ZINC250K-MolFlow, QM9-HierVAE, and ChEMBL-HierVAE, respectively. Most of the fragments do not show up or with very few times (less than 1\% occurrence frequency). Removing these fragments will lead to the following 11 motifs. (We have added these introduction in Sec C.1 with highlights.)
> - These 85 fragments cover the most common fragments, and this is sufficient as a primary step to explore the disentanglement of molecule DGMs.
>
>
>
> ### Experimental details
>
> Thank you for raising this question. We have moved more experimental details from appendix into the main manuscript in Sec 5. If you have any other concerns, feel free to explicitly point them out, and we are happy to provide more explanations.
>
>
> ### Accessibility for non-experts
>
> Thank you for raising this question. We are aware of this issue, and this is why we are marking different steerable factors with different colors in Fig 3. We have now highlighted them in Figure 4 caption.
> - In Fig 3.a, the halogens are marked in green, and you can see how the number of halogens decreses from [-3, 3].
> - In Fig 3.b, the hydroxyls are marked in red, and you can see how the number of hydroxyls decreses from [-3, 3].
> - In Fig 3.c, the amides are marked in red and blue, and you can see how the number of amides increase from [-3, 3].
> - In Fig 3.d, the chains are marked in green, and you can see how the number of chains is increased from [-3, 3].

---

> ### Author Response · Authors · 2023-11-13
> **Reply (2/2)**
>
> ### Additional questions
> 1. We provide the quantative stability and robustness of the key hyperparameters ($c_1, c_2, c_3$) in Sec F.3 and F.4.
> 2.
>     - First we want to clarify that in GraphCG, the representation and the representation space are fixed (please check the problem definition in Sec 2 and algorithm in Sec 4).
>     - Then about the question on the *failure* of the learned semantic directions. In the current experiment, we have fixed the number of 10 (as shown in the hyperparameter table in Sec E), and we observe semantic meanings on 4 of them. For the rest 6 directions, there are two possible reasons for why we didn't observe any semantic information yet: (1) The learned semantic directions do not cover any information; or (2) they do include semantic information, but we haven't discovered them yet. Recall that we are using post-training human selection for detecting the semantics. We also want to point out that this is still an open question in the general unsupervised editing ML community.
> 3. We have now added explicit definition of `sequences of edited graphs` in Sec 2. You can find the visualizations in Figure 1,3,4, where each point at each edited sequence correspond to a different step size. For instance for the edited point clouds in Figure 4, you can find two sequences with 5 edited graphs in Figure 4(a), 4(b), 4(c), and 4(d), respectively.
> 4. There are fundamental difference between the two methods, and we have provided detailed discussion in Sec 4.4. Here we also provide a brief summary of them:
>     - The optimized targets are different. SSL targets at pretraining a representation. In GraphCG (Eq 2), the encoder and decoder are fixed, i.e., the representation is fixed. What GraphCG learns is the semantic direction.
>     - The contrastive views are different. SSL pretrains the data representation at an inter-data level (between data points), while GraphCG learns the semantic directions at an inter-direction level (between semantic directions).

---

### Decision · Action_Editor_2mgh · 2024-01-24

**Recommendation:** Accept with minor revision

**Comment:**

The reviewers agree that the paper is worthy of publication in TMLR. Their main concerns regarding clarity were mostly addressed by the authors.

Here are som comments that should be addressed in a minor revision:
- it would be helpful to point out earlier on in Section 4 that the method relies on *positive* pairs z^u, z^v -- this is deferred to section 4.3, but it would be helpful to clarify this earlier.
- as pointed out by reviewer dKVp, it is not directly clear from the MI motivation in Section 4.1 how independence/disentanglement is enforced. In particular, the MI objective in eq.(5) seems to only consider two views from the *same* editing direction, while the repulsion between different direction i and j seems to first appear in the contrastive objective (9). Perhaps this comes from the fact that $i$ itself is randomly chosen, but is always the same in positive pairs? Please clarify this.
- In algorithm 1, steps 3 and 5 seem redundant?

**Audience:**

The paper and proposed method should be of interest to the community working on graph generation, as well as disentangled representations.

**Claims And Evidence:**

The paper introduces methods for learning semantically meaningful / disentangled directions in latent space of DGMs that may then be used for steerable generation (here in the context of graphs or point clouds). The reviewers found the method interesting and useful, and generally agreed that the claims are accurate, convincing, and the evidence clear. There are some concerns regarding the theory based on mutual information that should be addressed in a minor revision (see comments below).

---

> ### Author Response · Authors · 2024-01-29
> **Thank you for valuable comments**
>
> Dear Action Editor,
>
> Thank you for recognizing our work as accurate, convincing, and clear. We also appreciate you providing valuable comments! We have revised them carefully in the camera-ready version, and also briefly here:
>
> - Thank you for your comment. We have added this on Page 5, `Such two latent codes will be treated as positive pairs, and their construction will be introduced in Section 4.3.`
> - Thank you for raising this concern.
>     - First, we want to clarify is that the deep generative models (DGMs) are fixed, and we are not forcing this backbone model. Now we explained this further on page 5: `Recall that our ultimate goal is to enable graph editing based on semantic vectors. Existing deep generative models are entangled, thus obtaining such semantic vectors is a nontrivial task. To handle this problem, we propose using mutual information (MI) to learn the semantics. `
>     - About Eq 5, you are right that we only include positive pairs with semantic direction $i$. The direction $j$ comes out because we are using the noise contrastive estimation (NCE) to estimate the two conditional distributions in Eq 5 (Sec 4.2 and 4.3). This holds for arbitrary semantic direction $i$, which we will learn through Eq 11.
> - Thank you for checking our manuscript carefully. They are redundant, and we have merged step 5 into step 3 now.
>
> Authors of GraphCG